# Unveiling the influence of tumor and immune signatures on immune checkpoint therapy in advanced lung cancer

Nayoung Kim[1,2†], Sehhoon Park[3†], Areum Jo[1,2], Hye Hyeon Eum[1,2], Hong Kwan Kim[4], Kyungjong Lee[5], Jong Ho Cho[4], Bo Mi Ku[6], Hyun Ae Jung[3], Jong-Mu Sun[3], Se-Hoon Lee[3], Jin Seok Ahn[3], Jung-Il Lee[7], Jung Won Choi[7], Dasom Jeong[1,2], Minsu Na[1,2], Huiram Kang[1,2], Jeong Yeon Kim[8], Jung Kyoon Choi[8], Hae-Ock Lee[1,2,9*], Myung-Ju Ahn[3*]

[1]Department of Microbiology, College of Medicine, The Catholic University of Korea, Seoul, Republic of Korea; [2]Department of Biomedicine and Health Sciences, Graduate School, The Catholic University of Korea, Seoul, Republic of Korea; [3]Division of Haematology-Oncology, Department of Medicine, Samsung Medical Center, Sungkyunkwan University School of Medicine, Seoul, Republic of Korea; [4]Department of Thoracic and Cardiovascular Surgery, Samsung Medical Center, Sungkyunkwan University School of Medicine, Seoul, Republic of Korea; [5]Division of Pulmonary and Critical Care Medicine, Department of Medicine, Samsung Medical Center, Sungkyunkwan University School of Medicine, Seoul, Republic of Korea; [6]Research Institute for Future Medicine, Samsung Medical Center, Sungkyunkwan University School of Medicine, Seoul, Republic of Korea; [7]Department of Neurosurgery, Samsung Medical Center, Sungkyunkwan University School of Medicine, Seoul, Republic of Korea; [8]Department of Bio and Brain Engineering, KAIST, Daejeon, Republic of Korea; [9]Precision Medicine Research Center, College of Medicine, The Catholic University of Korea, Seoul, Republic of Korea

*For correspondence:
haeocklee@catholic.ac.kr (H-OckL);
silkahn@skku.edu (M-JuA)

†These authors contributed equally to this work

## eLife Assessment

The authors utilized single-cell RNA-seq profiling of non-small cell lung cancer (NSCLC) patient tumor samples to generate **useful** insights into the determinants of immune checkpoint inhibitor (ICI) responsiveness in NSCLC patients. While some of the findings add weight to the current literature, the analysis is **incomplete** due to the small cohort size and heterogeneous population which has limited their ability to draw statistically supported conclusion after adjusting for multiple hypothesis testing, as well as the lack of functional characterization of the findings. This study would benefit from external cohorts to both validate the findings and justify the statistical analysis undertaken.

**Abstract** This study investigates the variability among patients with non-small cell lung cancer (NSCLC) in their responses to immune checkpoint inhibitors (ICIs). Recognizing that patients with advanced-stage NSCLC rarely qualify for surgical interventions, it becomes crucial to identify biomarkers that influence responses to ICI therapy. We conducted an analysis of single-cell transcriptomes from 33 lung cancer biopsy samples, with a particular focus on 14 core samples taken before the initiation of palliative ICI treatment. Our objective was to link tumor and immune cell profiles with patient responses to ICI. We discovered that ICI non-responders exhibited a higher presence of CD4+ regulatory T cells, resident memory T cells, and TH17 cells. This contrasts with the diverse

activated CD8+ T cells found in responders. Furthermore, tumor cells in non-responders frequently showed heightened transcriptional activity in the NF-kB and STAT3 pathways, suggesting a potential inherent resistance to ICI therapy. Through the integration of immune cell profiles and tumor molecular signatures, we achieved an discriminative power (area under the curve [AUC]) exceeding 95% in identifying patient responses to ICI treatment. These results underscore the crucial importance of the interplay between tumor and immune microenvironment, including within metastatic sites, in affecting the effectiveness of ICIs in NSCLC.

## Introduction

Treatment landscape in cancer has rapidly evolved with the introduction of immune checkpoint inhibitors (ICIs). Among various immune checkpoints, antibody-targeting programmed cell death-1 (PD-1) and its ligand (PD-L1) have demonstrated clinical benefits over conventional systemic chemotherapy in patients with non-small cell lung cancer (NSCLC). They have been approved as either monotherapy in patients with high PD-L1 expression (*Reck et al., 2016*) or combined with cytotoxic chemotherapy regardless of PD-L1 expression (*Gandhi et al., 2018*; *Paz-Ares et al., 2018*). Moreover, the clinical benefits were validated in unresectable stage III NSCLC as consolidation therapy after definitive chemoradiotherapy or early-stage NSCLC (Ib-IIIA) as adjuvant therapy after curative surgery (*Wakelee et al., 2021*).

There have been many efforts to elucidate the predictive biomarkers of PD-(L)1 inhibitors. The PD-L1 expression in tumor tissue evaluated via immunohistochemistry has been incorporated as a companion diagnostic biomarker from the early clinical trials, which enhanced the response rate up to 46% in patients with PD-L1 ≥50% and showed an overall survival rate of up to 26.3 months (*Reck et al., 2021*). Other biomarkers, such as tumor mutation burden or gene expression profile, also demonstrated a positive predictive value (*Fehrenbacher et al., 2016*; *Hellmann et al., 2018*). Recent large-scale meta-analysis of clinico-immunogenomics shows that both tumor- and T cell-intrinsic factors exert a substantial impact on ICI response (*Litchfield et al., 2021*), supporting the necessity of in-depth investigation of high-throughput profiles of tumor and microenvironment.

ICI treatment modifies systemic immune responses, which can be monitored by alterations in the proportion of specific immune cell populations. An increase in PD-1+Ki67+CD8+ T cells in peripheral blood after PD-1 inhibitor treatment is associated with better outcomes in patients with NSCLC (*Kamphorst et al., 2017*; *Kim et al., 2019*). This cell type has also been identified in tumor tissues prior to the ICI treatment, where it is linked to clinical outcomes, and shows impaired production of classical effector cytokines (*Thommen et al., 2018*). More specifically, PD-1 expression is upregulated upon antigen recognition (*Simon and Labarriere, 2017*), indicating that certain T cells in the tumor microenvironment are actively engaged as tumor-specific T cells. Beyond CD8+ T cells, other immune cell types, such as myeloid-derived suppressor cells or regulatory T cells, which regulate tumor-specific T cell immunity, may also influence therapeutic outcomes (*Arce Vargas et al., 2017*; *Krieg et al., 2018*; *Kumagai et al., 2020*). Overall, the multicellular regulation of the tumor-immune microenvironment emphasizes the importance of profiling systemic tumor and immune cells at single-cell resolution to investigate factors associated with responses to ICI treatment.

## Results

### Variability and features of the lung cancer samples

We conducted single-cell RNA sequencing (scRNA-seq) on 33 lung cancer samples from 26 patients treated with ICIs between August 2017 and December 2019 to understand how cellular dynamics in lung cancer affect treatment sensitivity to PD-(L)1 inhibitors, used alone or in combination (*Figure 1a* and *Supplementary file 1*). Immune checkpoint therapy provides clinical benefit in advanced metastatic NSCLC across different treatment lines (*Ruiz-Patiño et al., 2020*). Notably, our samples have been collected from various tissue sites. In the scRNA-seq analysis, all specimens were used for the cell-type profiling in an unbiased manner. For the evaluation of clinical outcomes, only refined 14 core samples from 11 patients were used to minimize sample specific variations. Exclusion criteria from the core group encompass samples with treatment applied as adjuvant therapy, acquired after ICI treatment, no tumor content, non-evaluable for the clinical response, or histology other than NSCLC

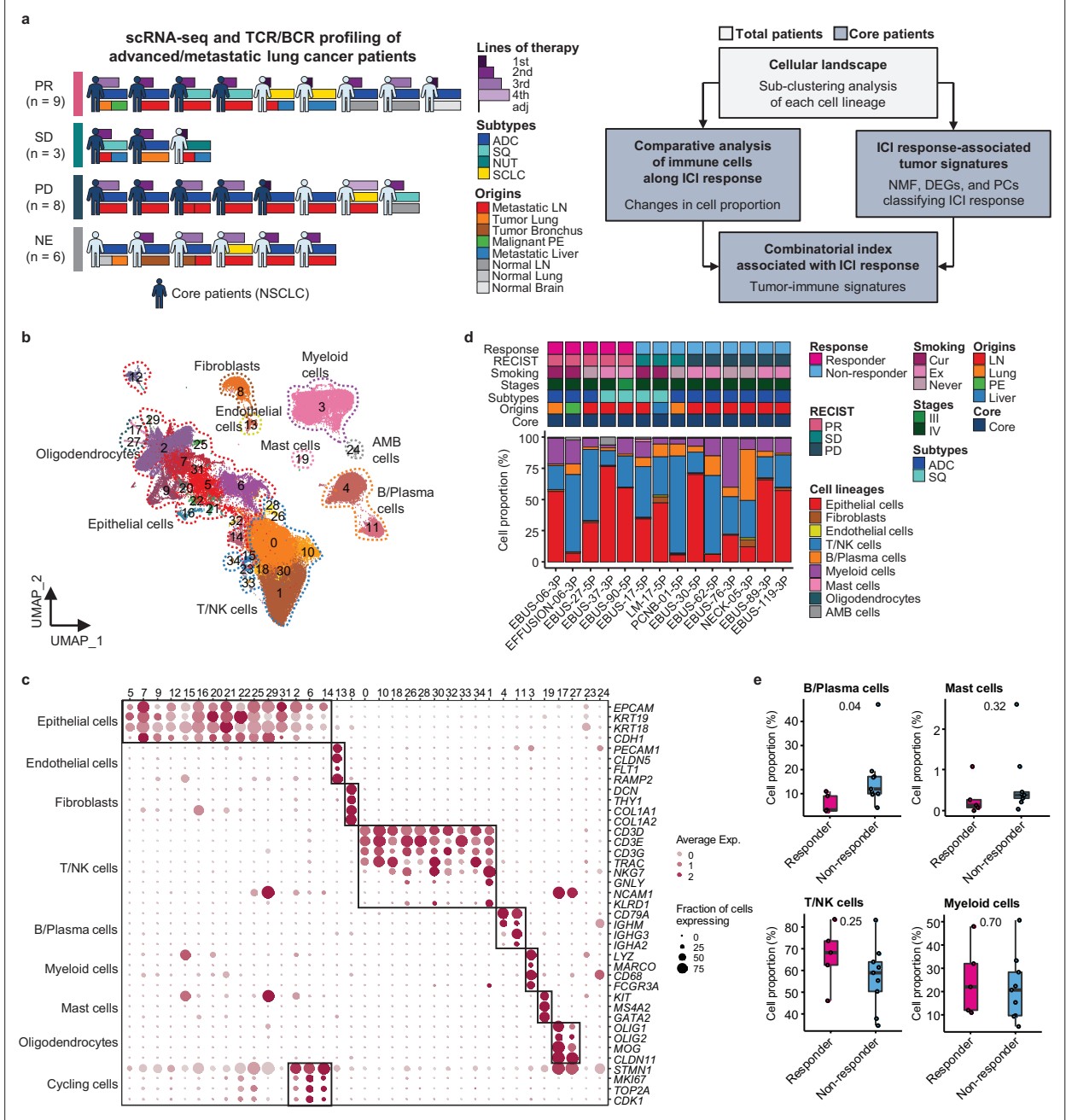

**Figure 1.** Cell lineage identification of 96,505 single cells from 26 patients with lung cancer treated with immune checkpoint inhibitor (ICI). (**a**) Workflow of sample collection and single-cell analysis of lung cancer patients treated with ICI. (**b**) Uniform Manifold Approximation and Projection (UMAP) plot of 96,505 single cells from 33 samples acquired from 26 advanced lung cancer patients, colored by clusters. AMB cells, ambiguous cells. (**c**) Dot plot of mean expression of canonical marker genes for cell lineages. (**d**) Proportions of the cell lineages in non-small cell lung cancer (NSCLC) tissue from core patients shown by individual samples aligned with clinical data. Labels for origins indicate LN, metastatic lymph node; Lung, tumor lung; PE, malignant pleural effusion; Liver, metastatic liver. (**e**) Box plot of the percentage of cell lineages in responder and non-responder groups. Label represents p-value calculated via two-tailed Student's t-test. Each box represents the median and the interquartile range (IQR, the range between the 25th and 75th percentile), whiskers indicate the 1.5 times of IQR.

The online version of this article includes the following figure supplement(s) for figure 1:

**Figure supplement 1.** Immune cell profiles for sample collection sites in lung adenocarcinoma (LUAD).

**Table 1.** Clinical overview of non-small cell lung cancer (NSCLC) patients treated with immune checkpoint inhibitor (ICI).

| Immunotherapy | Targets | No. of samples | No. of patients |
|---|---|---|---|
| Pembrolizumab | PD-1 | 9 | 8 |
| Atezolizumab | PD-L1 | 1 | 1 |
| Nivolumab | PD-1 | 2 | 1 |
| Vibostolimab+Pembrolizumab | TIGIT+PD-1 | 2 | 1 |
| **RECIST** | **Description** | **No. of samples** | **No. of patients** |
| PR | Partial response | 5 | 4 |
| SD | Stable disease | 3 | 2 |
| PD | Progressive disease | 6 | 5 |

such as nuclear protein in testis (NUT) and small cell lung cancer (*Table 1* and *Figure 1a*). Of the 11 core patients, 8 had adenocarcinoma and 3 had squamous cell carcinoma (SQ). Clinical outcomes of ICI were partial response (PR) in four patients, stable disease (SD) in two patients, and progressive disease (PD) in five patients. Patients were classified as responders (PR) and non-responders (SD and PD) according to ICI response.

Due to the diversity of sample collection sites, our data may be influenced by varied immune cell composition at different sample collection sites. Therefore, we analyzed immune and stromal cell subsets across early-stage (tLung) and late-stage (tL/B) lung tumors, and metastatic lymph nodes (mLN), comparing them to normal lung (nLung) and lymph node (nLN) tissues. This analysis was conducted on public scRNA-seq data from 43 samples from 33 lung adenocarcinoma (LUAD) patients (*Kim et al., 2020*; *Figure 1—figure supplement 1a–c*). Although there were differences in tissue-specific resident populations, we found that the immune cell profiles, especially T/NK cells of mLN, were similar to those of primary tumor tissues (*Figure 1—figure supplement 1d–g*).

## Classification of immune cell subset in lung cancer

Global cell-type profiling (*Figure 1b and c* and *Supplementary file 2*) illustrates the cellular composition of each sample as epithelial/tumor cells, fibroblasts, endothelial cells, T/natural killer (NK) cells, B/plasma cells, myeloid immune cells, and mast cells. Individual samples show variations in epithelial/tumor content as well as in immune cell composition (*Figure 1d and e*).

For further analysis of immune cell subtypes, we applied sequential subclustering on global immune cell clusters. As scRNA-seq shows limited performance in separating CD4+ and CD8+ T cell subsets, antibody-derived tag (ADT) information (*Stoeckius et al., 2017*) was used to complement the transcriptome data and to predict CD4+ T cells, CD8+ T cells, and NK cells (*Figure 2a*). Finally, 14 CD4+ and 14 CD8+ T cell subclusters were identified excluding <5% ambiguous cells (*Figure 2b and c* and *Supplementary file 2*). In the CD4+ T cell compartment, naïve-like T cells (TN, CD4_cluster0) and central memory T cells (TCM, CD4_cluster1) expressing *SELL, TCF7, LEF1,* and *CCR7* genes or tissue-resident memory T cells (TRM, CD4_clusters3, 5, 6) expressing *NR4A1, MYADM,* and *PTGER4* genes were abundant in most samples. Regulatory T cells (Tregs, CD4_cluster2) with *FOXP3, CTLA4, ICOS,* and *BATF* expression were also abundant, which has been demonstrated as tumor-specific alterations in the tissue microenvironment (*Kim et al., 2020*; *Guo et al., 2018*; *Gueguen et al., 2021*). In the CD8+ T cell compartment, effector memory T cells (TEM), effector T cells (TEFF), effector memory CD45RA positive cells (TEMRA) (CD8_clusters0, 2, 3, 8) expressing *PRF1* and *IFNG* were dominant over TN/TCM (CD8_clusters4, 5) types. Exhausted T cells (TEX, CD8_clusters1, 12) expressed multiple checkpoint genes (*HAVCR2* and *PDCD1*) along with high levels of *PRF1, IFNG, CXCR3,* and *CXCL13*. Co-expression of cytotoxic effectors and checkpoint molecules in TEX clusters indicates that cluster populations may retain functional capacity as cytotoxic TEFF (*Groom and Luster, 2011*). Further, clonotype analysis of T cell receptor (TCR) supported the T cell subset classification demonstrating higher clonal expansion in the CD8+ T cell compartment than that in CD4+ T cells, with the highest levels within the TEX subclass (*Figure 2—figure supplement 1*).

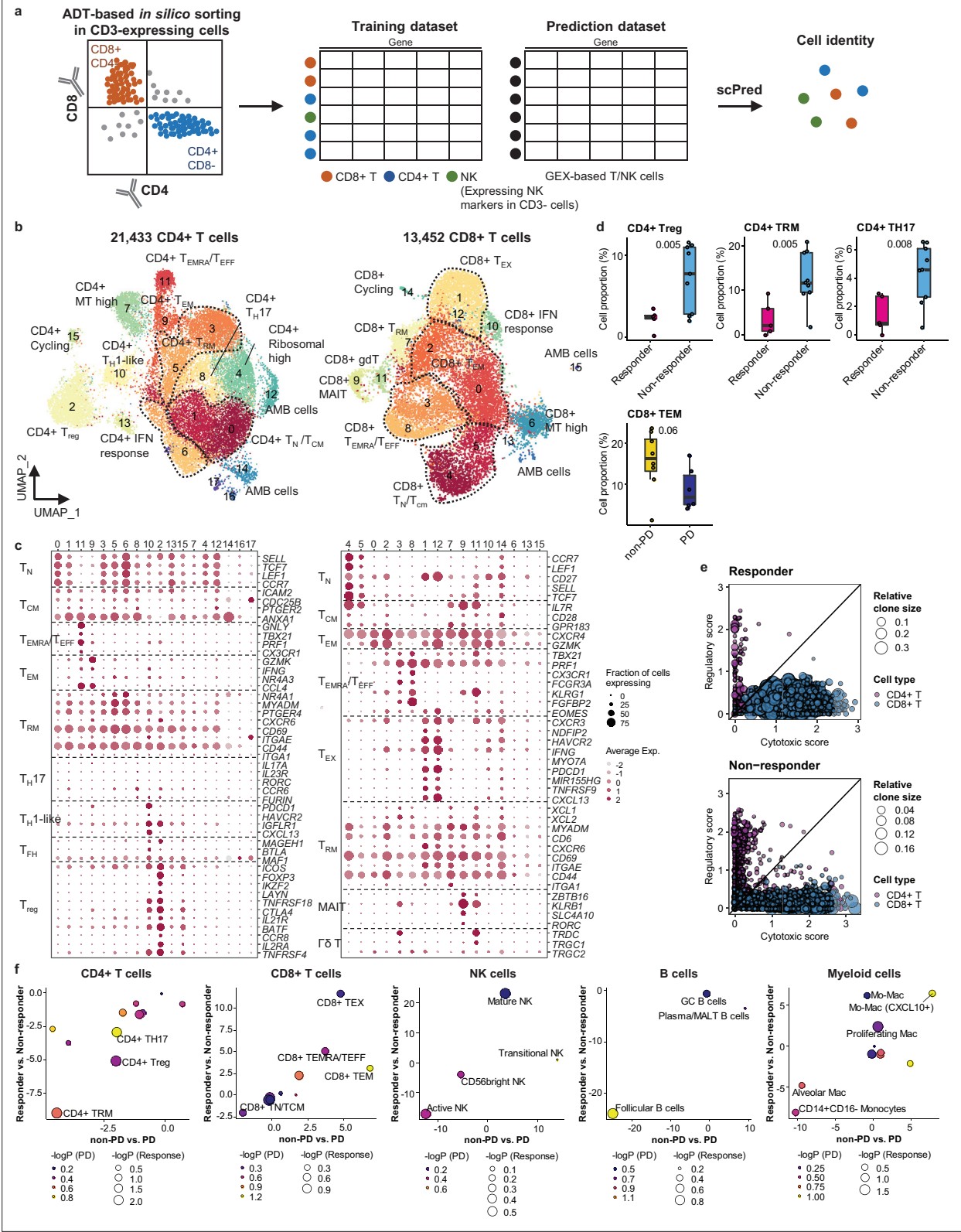

**Figure 2.** Classification and characterization of CD4+ and CD8+ T cell subtypes. (**a**) Prediction strategy to classify CD4+, CD8+ T, and natural killer (NK) cells by applying antibody-derived tag (ADT) data from lung cancer. (**b**) Uniform Manifold Approximation and Projection (UMAP) plot of CD4+ and CD8+ T cells, colored by clusters. (**c**) Dot plot of mean expression of selected CD4+ (left) and CD8+ (right) T cell marker genes in each cell cluster. (**d**) Box plot of the percentage of CD4+ and CD8+ T cell types within total CD4+ plus CD8+ cells for sample groups representing responses to immune

*Figure 2 continued on next page*

*Figure 2 continued*

checkpoint inhibitor (ICI). Label represents p-value calculated via two-tailed Student's t-test. Each box represents the median and the interquartile range (IQR), whiskers indicate the 1.5 times of IQR. (**e**) Association of T cell functional features with clonal expansion. Dot size depicts the relative clone size of each cell, which is divided by the total number of CD4+ and CD8+ T cells, respectively. Color indicates the cell lineage. (**f**) Comparisons of proportional changes in cell subtypes along ICI responses within each immune cell lineage. The quantitative values shown on the axis represent the mean difference in % cell proportions between sample groups. Dot size and color represent –log (p-value) for responder vs. non-responder and non-progressive disease (PD) vs. PD, respectively. The lower left quadrant shows cell types overrepresented in the poor responder groups, while the upper right quadrant indicates cell types overrepresented in the better responder groups. p-Value, two-tailed Student's t-test.

The online version of this article includes the following figure supplement(s) for figure 2:

**Figure supplement 1.** Features of T cell receptor (TCR) repertoires of CD4+ and CD8+ T cells.

**Figure supplement 2.** Heterogeneity in natural killer (NK), B, and myeloid cells and features of B cell receptor (BCR) repertoires.

NK cells can be subclassified as CD56bright, transitional, active, and mature types (*Yang et al., 2019*; *Figure 2—figure supplement 2a and b* and *Supplementary file 2*). Active NK cells expressed the highest level of *PRF1*, *TNF*, and *IFNG*, reflecting a cytotoxic effector function.

Compared to T cell clusters, fewer B/plasma cells were detected as follicular (B_clusters 0, 1, 2, 4, 6, 11), germinal center (B_cluster 10), and plasma/mucosa-associated lymphoid tissue (MALT) B cells (B_clusters 3, 7, 12) (*Figure 2—figure supplement 2a and b* and *Supplementary file 2*). Plasma/MALT B cells manifested higher levels of clonal expansion of B cell receptor (BCR) than follicular B cells (*Figure 2—figure supplement 2c and d*). Identical clonality of some follicular B and plasma cells suggests in situ maturation and differentiation of B cells to plasma cells in tumor tissues (*Figure 2—figure supplement 2c*, clonotype 16).

Myeloid cells were composed of monocytes, dendritic cells, and a large number of macrophages (*Figure 2—figure supplement 2a and b* and *Supplementary file 2*). CD14+CD16- classical monocytes (Myeloid_clusters1,6) were predominantly found over non-classical CD14$^{lo}$CD16+ types (Myeloid_cluster15). Alveolar macrophages (Alveolar Mac, Myeloid_cluster0) expressed well-defined marker genes such as *MARCO*, *FABP4*, and *MCEMP1* along with anti-inflammatory genes such as *CD163, APOE,* and *C1QA/B/C*. Monocyte-derived macrophages represent heterogeneous populations with a similar gene expression profile to alveolar macrophages (Mo-Mac, Myeloid_clusters2, 3, 5, 9, 11, 13), with an elevated chemokine gene expression (CXCL10+ Mo-Mac, Myeloid_clusters4, 14, 16), or with active cell-cycle progression (Proliferating Mac, Myeloid_cluster7). Dendritic cells were categorized as CD1c+ (Myeloid_cluster8, *CD1C* and *ITGAX*), activated (Myeloid_cluster17, *CCR7* and *LAMP3*), and CD141+ (Myeloid_cluster19, *CLEC9A* and *XCR1*) subclusters.

Overall immune cell composition is comparable to those reported in previous studies (*Kim et al., 2020*; *Guo et al., 2018*; *Gueguen et al., 2021*).

## Immune cell landscape fostering the ICI response

In global cell-type profiling (*Figure 1b and c*), abundance in T, NK, or myeloid cell types shows no difference between responders and non-responders (*Figure 1d and e*). Nonetheless, as specific differentiation features within the cell types may influence the response to ICI treatment, we compared the response groups using the proportion of subclusters within CD4+ T, CD8+ T, NK, B/plasma, and myeloid cells. After subclustering (*Figure 2b* and *Figure 2—figure supplement 2a*), three subsets of CD4+ T cells, i.e., Treg, TRM, and CD4+ T helper 17 (TH17), were significantly (p<0.01) overrepresented in the non-responder group (*Figure 2d*). In contrast, among CD8+ T cell populations, TEM subsets demonstrated a modest level of association with the patients who responded well against PD (*Figure 2d*, p=0.06). TCR clonotype analysis supported the cellular dynamics such that clonal expansion was more prominent in cytotoxic CD8+ T cells over CD4+ Tregs in the responder group (*Figure 2e* and *Figure 2—figure supplement 1c*). Overall landscape in each cell type (*Figure 2f*) suggests that CD4+ Treg and TRM as well as follicular B cells may interfere with the ICI response, whereas CD8+ T cell activation (TEM, TEMRA/TEFF, and TEX), mature NK cells, and CXCL10+ Mo-Mac cells support the ICI response. The balance between separate immune cell types informs immune regulatory axes that may be targeted to favor the activation of tumor-reactive immunity.

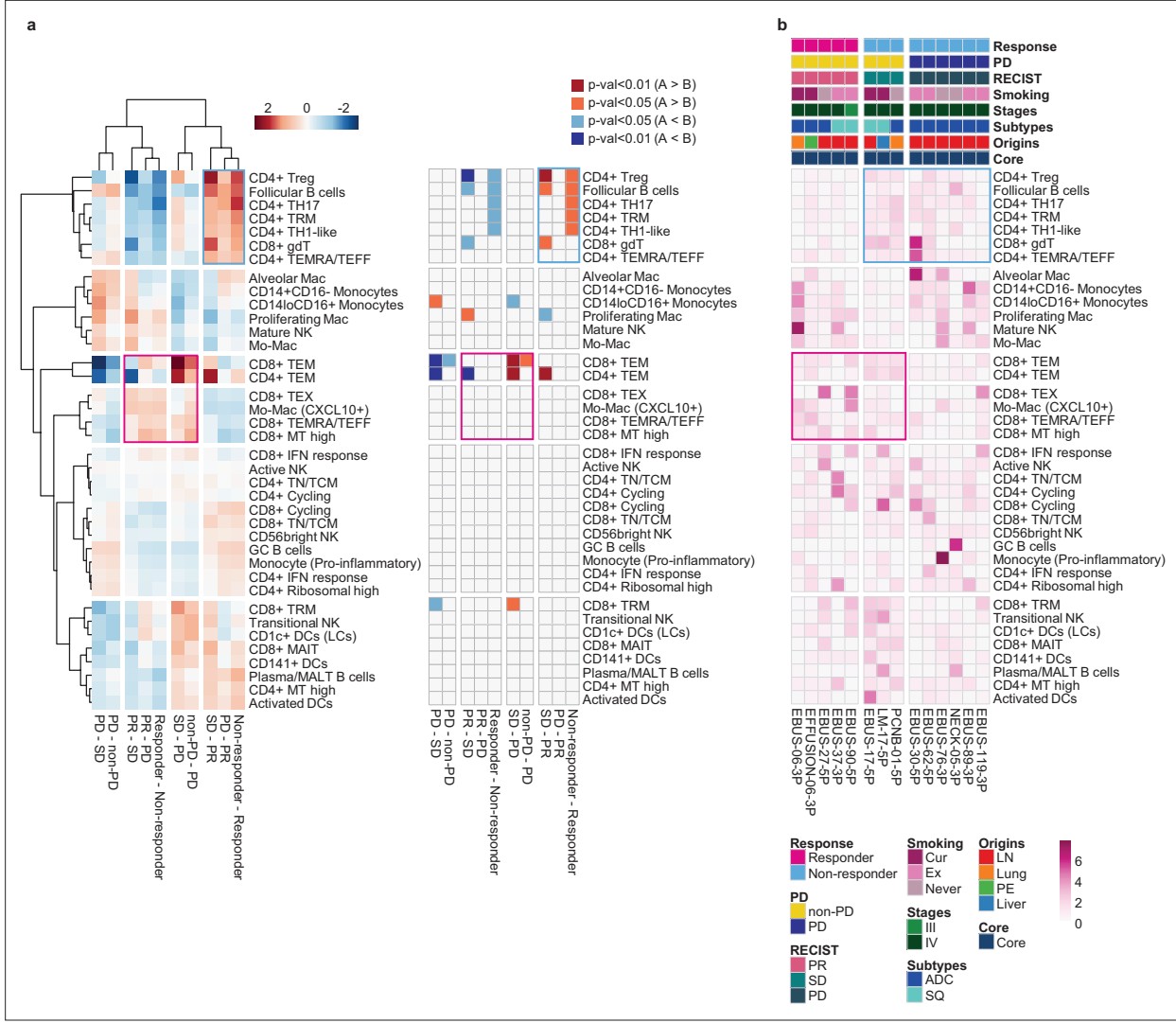

**Figure 3.** Systemic evaluation of immune cell dynamics associated with response to immune checkpoint inhibitor (ICI). (**a**) Heat map with unsupervised hierarchical clustering (left) and depicting significance (right) of proportional changes in cell subtypes within total immune cells. Proportional changes were compared for multiple ICI response groups. Color represents the –log (p-value) determined using two-tailed Student's t-test. (**b**) Distribution map for each cell type across individual samples aligned with clinical data. Color represents *Ro/e* score calculated using the chi-square test.

## Systemic evaluation of the immune microenvironment associated with ICI response

Next, we evaluated the immune microenvironment as an entity by using all immune cells as a denominator in the subtype proportions. In this setting, we used diverse clinical group comparisons and identified the immune cell blocks separated by clinical outcomes (*Figure 3a*). The immune cell blocks overrepresented in the non-responder groups consisted of CD4+ Treg, follicular B cells, and CD4+ TH17/TRM/T helper 1 (TH1)-like cells. In the immune cell blocks of the responder groups, CD8+ TEM cells showed the strongest enrichment along with the other CD8+ TEX/TEMRA/TEFF/mitochondria (MT) high cells as well as CXCL10+ Mo-Mac. Despite the immune footprints of the ICI responders, extensive variations among individual patients (*Figure 3b*) hamper patient stratification solely based on the immune profiles.

## Tumor cell signatures associated with ICI response

We investigated the associations between genomic characteristics in tumor and ICI response. The constraints of mutation analysis with 10x chromium data complicated the direct correlation between

tumor mutation burden and ICI outcome. Rather, we assessed copy number alterations (CNA) indirectly, via chromosomal gene expression patterns. These analyses revealed a moderate correlation between low levels of CNA, including both gain and loss of heterozygosity, and positive responses to ICI (*Figure 4—figure supplement 1*). This finding is consistent with the result from previous genetic association studies (*Liu et al., 2019*).

To assess gene expression characteristics of tumors influencing ICI response, we separated malignant tumor cell clusters from normal epithelial cell types (*Figure 4—figure supplement 2*). Subsequent differentially expressed gene (DEG) analysis (*Figure 4a* and *Supplementary file 3*) identified genes in poor response groups linked to the regulation of cell death, cell motility, and cell activation (*Figure 4—figure supplement 3* and *Supplementary file 4*). The DEGs were refined later by combinations of various tumor signatures separating responder and non-responder groups (*Supplementary file 5*).

Next, to explore the existence of gene programs and modules influencing the ICI response, we applied factorization using non-negative matrix factorization (NMF) and scINSIGHT (*Qian et al., 2022*). Among 30 factors from NMF across all malignant cells, we identified factors showing high loadings for a specific Response Evaluation Criteria in Solid Tumor (RECIST) group as NMF programs p1~4 (*Figure 4b and c*). There were clear distinction among RECIST groups according to the gene expression levels associated with these NMF programs (*Figure 4d* and *Supplementary file 5*). To identify gene modules consistent across different patients, we examined RECIST-specific modules though scINSIGHT analysis (*Qian et al., 2022*; *Figure 4e*). Unfortunately, we found that contributions to these gene modules varied significantly among patients. To mitigate this variability, we adjusted the RECIST-specific modules by combining genes from the original modules (*Supplementary file 5*). The refined gene modules showed a specific gene expression pattern for each RECIST group, similar to the NMF programs (*Figure 4f*). Overall, both genes and their functional categories segregated depending on the selection techniques used (*Figure 4g and h*). However, transcription factors governing the signatures derived from DEG, NMF, and scINSIGHT analyses consistently delineated between responders and non-responders (*Figure 4i*). Responder-specific gene signatures showed associations with the transcription factors Regulatory Factor X Associated Ankyrin Containing Protein (RFXANK), Regulatory Factor X Associated Protein (RFXAP), and Regulatory Factor X5 (RFX5). This RFX protein complex has emerged as a positive biomarker for the immune response in diverse cancer types (*Lapuente-Santana et al., 2021*). Non-responder-specific gene signatures were regulated by Activator Of Transcription 3 (STAT3) and Nuclear Factor Kappa B Subunit 1 (NFKB1), known to play roles in PD-L1 regulation and T cell activation in cancer (*Betzler et al., 2020*).

We also adopted principal component analysis (PCA) to isolate correlated gene signatures variably expressed in tumor cells (*Figure 4—figure supplement 4* and *Supplementary files 4 and 5*). Among the top 10 PCs, negatively correlated genes in PC2, PC7, and PC8 distinguished the tumor cells in the poor response groups, whereas positively correlated genes in PC6 and PC9 were upregulated in the better response groups (*Figure 4—figure supplement 5a–c*). Tumor cells from the PR group had low PC2.neg scores, suggesting low growth factor/type I interferon response signaling as a tumor cell-specific positive predictor of the ICI response. Conversely, high levels of growth factor/type I interferon response signaling in tumor cells may present intrinsic resistance to PD-(L)1 inhibitor alone or in combination. Type I interferon is known to drive anti-tumor effect directly or indirectly on tumor and surrounding immune cells, but also acts to counter the anti-tumor effect by inducing CD8+ T cell exhaustion and upregulating immune-suppressive genes on tumor cells (*Fenton et al., 2021*). We assessed whether tumor cell signatures are applicable in association with ICI response in other tumors. They had a modest influence on the response to ICI treatment of melanoma (*Figure 4—figure supplement 5d*) in bulk gene expression data (*Van Allen et al., 2015*; *Riaz et al., 2017*).

These tumor cell signatures were specifically linked to elucidating the ICI response, but did not demonstrate any prognostic value for LUAD or lung squamous cell carcinoma (LUSC) in the The Cancer Genome Atlas (TCGA) RNA sequencing data (*Table 2*).

## Combination of tumor signatures and immune cell dynamics classify ICI response

Immune cell dynamics or tumor signature alone has a limited capacity to profile the therapeutic outcome of PD-(L)1 inhibitor alone or in combination (*Figure 5a*). Similar to the immune cell blocks

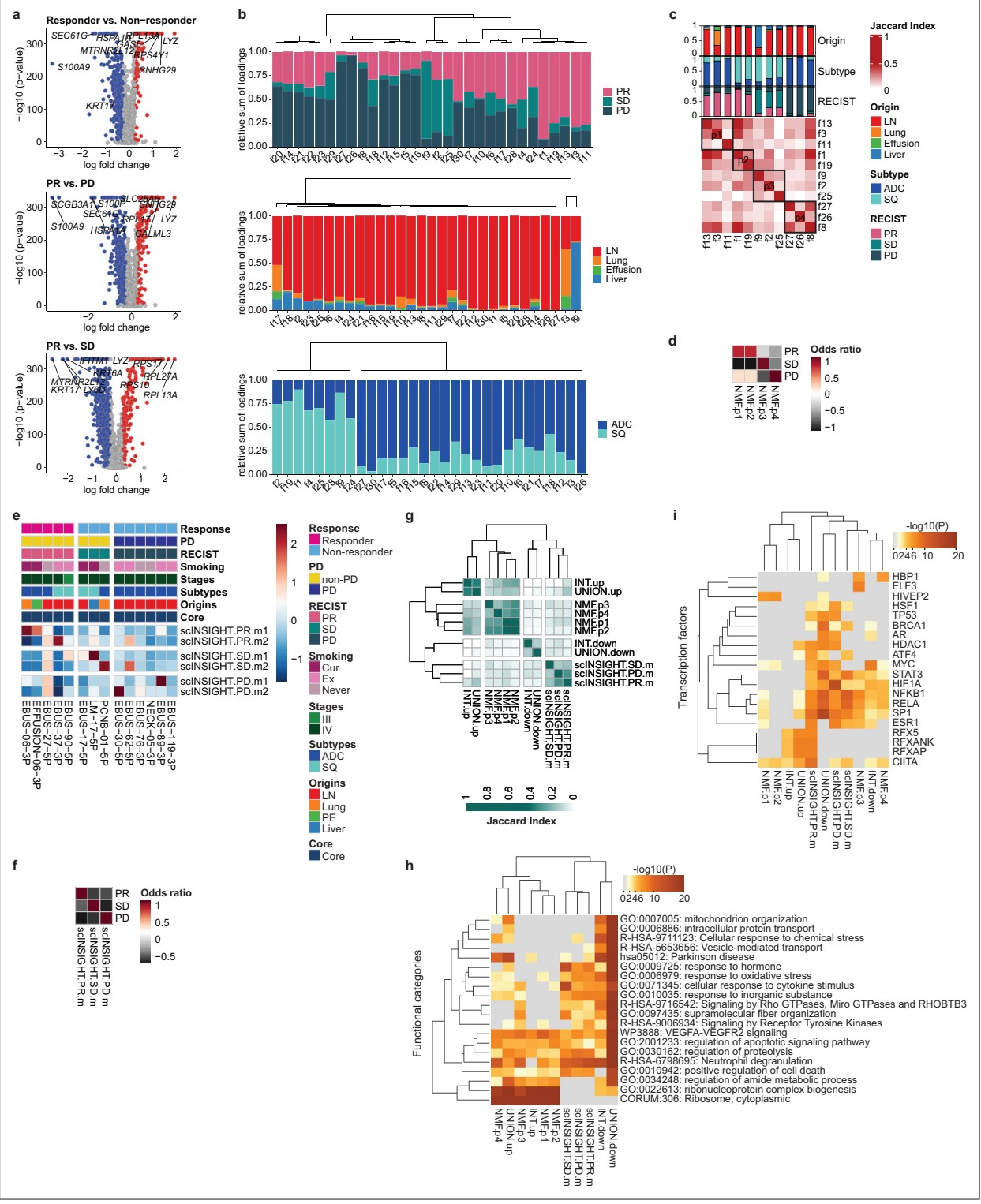

**Figure 4.** Single-cell tumor signatures associated with response to immune checkpoint inhibitor (ICI). (**a**) Volcano plot of expression difference for responder vs. non-responder, partial response (PR) vs. progressive disease (PD), and PR vs. stable disease (SD) in 12,975 malignant cells from 11 core patients. The log fold change indicates the difference in the mean expression level for each gene. The significance level was determined using two-tailed Wilcoxon rank sum test. (**b**) Relative sum of loadings for all non-negative matrix factorization (NMF) factors contributed to malignant cells from 11 core patients across Response Evaluation Criteria in Solid Tumor (RECIST), tissue origins, and cancer subtypes, respectively. (**c**) Selection of RECIST-enriched NMF programs. (**d**) Enrichment of NMF programs for RECIST groups. Color represents the z-transformed odds ratio. (**e**) Expression map of RECIST-specific scINSIGHT modules across individual samples aligned with clinical data. Color represents the z-transformed mean expression of genes

*Figure 4 continued on next page*

*Figure 4 continued*

contributing to each module. (**f**) Enrichment of RECIST-specific gene modules for RECIST groups. Color represents the z-transformed odds ratio. (**g**) Hierarchical clustering of pairwise similarities between tumor signatures. INT and UNION, intersection and union of differentially expressed genes (DEGs) for responder vs. non-responder, PR vs. PD, and PR vs. SD in (**a**). (**h**) Functional categories and (**i**) transcription factors of the selected tumor signatures, analyzed by Metascape.

The online version of this article includes the following figure supplement(s) for figure 4:

**Figure supplement 1.** Copy number alteration (CNA) profiles in non-small cell lung cancer (NSCLC) tissue aligned with immune checkpoint inhibitor (ICI) response.

**Figure supplement 2.** Classification of malignant cells based on genomic perturbations.

**Figure supplement 3.** Functional categories of single-cell differentially expressed gene (DEG) signatures associated with response to immune checkpoint inhibitor (ICI).

**Figure supplement 4.** Selection and characteristics of principal component (PC) signatures in malignant cells.

**Figure supplement 5.** Immune checkpoint inhibitor (ICI) response association of principal component (PC) signatures in malignant cells.

and tumor signatures that were overrepresented in the poor response group, each of CD4+ Treg, CD4+ TH17, PC7.neg, INT.down, and UNION.down was significantly associated with ICI response in univariate regression analysis (*Figure 5b*). PC7.neg denotes genes negatively correlated with PC7, a principal component (PC) extracted from PCA that distinguishes tumor cells in poor response groups. INT.down and UNION.down represent the intersection (INT) and union (UNION) of downregulated genes in the responder group, respectively. The variation in ICI response was not affected by clinical variables of tissue origin, cancer subtype, pathological stage, and smoking status. When we performed a combined analysis of the top tumor-immune features to classify response, the discriminative power (area under the curve [AUC]) was improved to over 95% (*Figure 5c*). Overall, features of the non-responders, especially CD4+ Treg, B/plasma cells, INT.down, and UNION.down, showed a higher estimate than those of responders. These non-responder features suggest heterogeneous mechanisms of resistance conferred by tumor and immune regulatory axes.

## Discussion

ICI alone, or in combination with chemotherapy, are considered standard first-line therapy for patients with NSCLC. NSCLC may harbor large numbers of genetic perturbations due to genotoxic environmental exposure, which likely generate high mutation burden or neoantigens (*Lawrence et al., 2013*). The neoantigen-directed T cell response is hampered by diverse immune suppressive mechanisms exerted by tumor cells and the immune regulatory network (*Sharma et al., 2017*). Current ICIs targeting PD-(L)1 aim one angle of many suppressive mechanisms, and identifying the features of non-responders will reveal additional regulatory angles to improve ICI response.

Immune regulatory network would determine the balance between activation and suppression of tumor-directed immunity. In previous studies, prediction of the response to ICI highlighted CD8+ cytotoxic TEFF and CD4+ Tregs (*Gibellini et al., 2020*). The involvement of effector CD8+ T cells is consistent in most studies regardless of the cellular origin (blood or tissues) or tumor type (melanoma, lung cancer) (*Zheng et al., 2021*). Their phenotypes slightly differ depending on the cellular resolution of the study. Our data provided the highest resolution cell types, and CD8+ TEM cells (*GZMK, CXCR4* expression) were overrepresented in the responders. By comparison, Tregs were underrepresented in the responder group. Previously, our group reported a contrasting result demonstrating an increase in Tregs in the posttreatment blood samples (not baseline) in the responder group (*Koh et al., 2020*). This discrepancy can be explained by the differences in the measurements, sites, and timing of sampling, and the resolution of subpopulations. In mouse preclinical models, PD-L1 inhibitor treatment induces T cell expansion of all phenotypes including CD4+/CD8+ TEFF and Tregs (*Wei et al., 2019*). Thus, Treg expansion captured in the posttreatment blood samples may represent overall immune activation in human patients. Alternatively, heterogeneity of Tregs and complex effects of PD-(L)1 inhibition on this cell type may contribute to variable results in response prediction.

The identification of an abundance of CD4+ TRM cells as a negative predictor of ICI response is an unexpected finding, considering that higher frequencies of TRM cells in lung tumor tissues are generally associated with better clinical outcomes in NSCLC (*Ganesan et al., 2017*). This is largely due

**Table 2.** Multivariate overall survival analysis of tumor signature genes.

| | TCGA LUAD | | | | TCGA LUSC | | | |
|---|---|---|---|---|---|---|---|---|
| | HR | Lower CI | Upper CI | p-Value | HR | Lower CI | Upper CI | p-Value |
| NMF.p1 | 0.89 | 0.53 | 1.48 | 0.64 | 0.92 | 0.66 | 1.27 | 0.60 |
| NMF.p2 | 0.87 | 0.53 | 1.43 | 0.59 | 0.93 | 0.67 | 1.28 | 0.66 |
| NMF.p3 | 0.76 | 0.49 | 1.21 | 0.25 | 1.06 | 0.76 | 1.49 | 0.72 |
| NMF.p4 | 1.00 | 0.64 | 1.55 | 0.98 | 1.00 | 0.70 | 1.44 | 0.98 |
| scINSIGHT.PR.m | 0.97 | 0.63 | 1.51 | 0.90 | 0.87 | 0.62 | 1.23 | 0.44 |
| scINSIGHT.SD.m | 0.81 | 0.53 | 1.23 | 0.32 | 0.94 | 0.67 | 1.33 | 0.74 |
| scINSIGHT.PD.m | 0.89 | 0.57 | 1.40 | 0.62 | 0.87 | 0.62 | 1.24 | 0.45 |
| PC1.pos | 0.48 | 0.27 | 0.84 | 0.01 | 0.78 | 0.55 | 1.12 | 0.17 |
| PC1.neg | 0.98 | 0.65 | 1.48 | 0.92 | 0.72 | 0.49 | 1.06 | 0.10 |
| PC2.pos | 0.87 | 0.55 | 1.36 | 0.53 | 0.98 | 0.69 | 1.40 | 0.91 |
| PC2.neg | 0.82 | 0.53 | 1.25 | 0.35 | 0.90 | 0.63 | 1.27 | 0.54 |
| PC3.pos | 1.14 | 0.68 | 1.90 | 0.63 | 0.87 | 0.63 | 1.19 | 0.38 |
| PC3.neg | 0.86 | 0.55 | 1.35 | 0.52 | 0.79 | 0.54 | 1.16 | 0.23 |
| PC4.pos | 0.88 | 0.54 | 1.42 | 0.59 | 0.66 | 0.46 | 0.95 | 0.02 |
| PC4.neg | 1.10 | 0.75 | 1.62 | 0.62 | 0.97 | 0.69 | 1.35 | 0.84 |
| PC5.pos | 1.09 | 0.71 | 1.68 | 0.68 | 0.76 | 0.53 | 1.10 | 0.15 |
| PC5.neg | 1.01 | 0.69 | 1.50 | 0.95 | 1.03 | 0.71 | 1.49 | 0.88 |
| PC6.pos | 0.77 | 0.45 | 1.29 | 0.32 | 1.09 | 0.79 | 1.51 | 0.60 |
| PC6.neg | 0.68 | 0.41 | 1.13 | 0.13 | 0.81 | 0.58 | 1.13 | 0.21 |
| PC7.pos | 1.10 | 0.71 | 1.72 | 0.67 | 0.75 | 0.51 | 1.10 | 0.14 |
| PC7.neg | 0.88 | 0.58 | 1.33 | 0.53 | 0.77 | 0.55 | 1.09 | 0.14 |
| PC8.pos | 0.77 | 0.50 | 1.18 | 0.23 | 1.01 | 0.72 | 1.43 | 0.94 |
| PC8.neg | 1.00 | 0.65 | 1.52 | 0.99 | 0.95 | 0.66 | 1.37 | 0.80 |
| PC9.pos | 0.91 | 0.60 | 1.38 | 0.66 | 1.02 | 0.73 | 1.43 | 0.91 |
| PC9.neg | 1.11 | 0.73 | 1.68 | 0.63 | 1.02 | 0.72 | 1.45 | 0.89 |
| PC10.pos | 1.13 | 0.76 | 1.68 | 0.55 | 1.28 | 0.88 | 1.86 | 0.20 |
| PC10.neg | 0.93 | 0.59 | 1.46 | 0.75 | 0.93 | 0.65 | 1.35 | 0.71 |
| INT.up | 0.82 | 0.53 | 1.26 | 0.37 | 1.00 | 0.71 | 1.42 | 0.99 |
| UNION.up | 0.80 | 0.52 | 1.21 | 0.28 | 0.77 | 0.53 | 1.13 | 0.18 |
| INT.down | 0.84 | 0.56 | 1.24 | 0.37 | 1.07 | 0.73 | 1.57 | 0.74 |
| UNION.down | 0.67 | 0.42 | 1.06 | 0.08 | 0.75 | 0.52 | 1.07 | 0.11 |

to their role in sustaining high densities of tumor-infiltrating lymphocytes and promoting anti-tumor responses. Additionally, previous studies have demonstrated that TRM cell subsets co-expressing PD-1 and TIM-3 are relatively enriched in patients who respond to PD-1 inhibitors (*Clarke et al., 2019*). However, recent findings suggest that pre-existing TRM-like cells in lung cancer may promote immune evasion mechanisms, contributing to resistance to immune checkpoint blockade therapies (*Weeden et al., 2023*). These observations suggest that the roles of TRM subsets in tumor immunity are highly context-dependent.

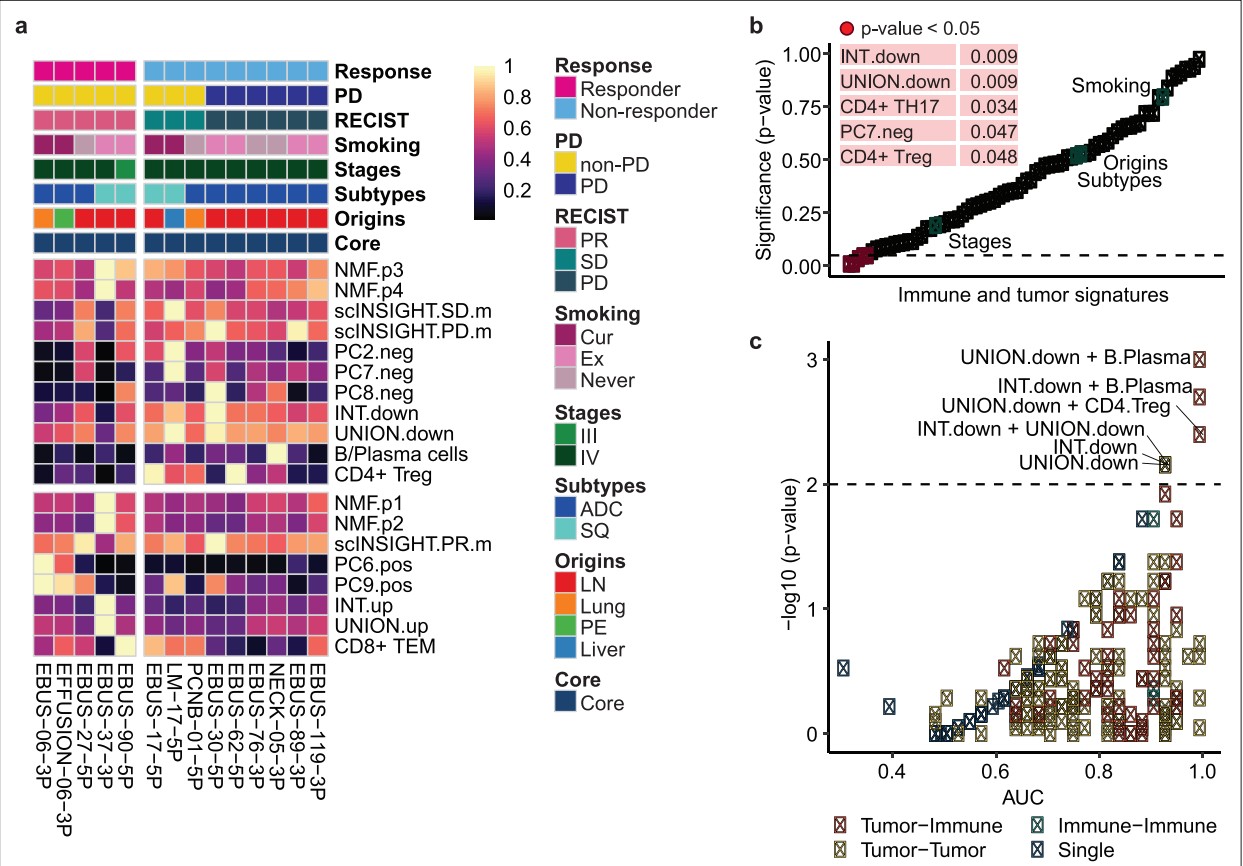

**Figure 5.** Combination of tumor signatures and immune index classifying the response to immune checkpoint inhibitor (ICI). (**a**) Heat map of relative contribution of tumor signatures and immune index across individual samples aligned with clinical data. Mean expression of each tumor signature and the percentage of each immune index are divided by the maximum value across samples. INT and UNION, intersection and union of differentially expressed genes (DEGs) for responder vs. non-responder, partial response (PR) vs. progressive disease (PD), and PR vs. stable disease (SD) in *Figure 4a*. (**b**) Univariate regression analysis of immune and tumor signatures for ICI response, together with clinical variables. (**c**) Receiver operating characteristic (ROC) analysis of combinatorial index to classify responder and non-responder. p-Value, two-tailed Wilcoxon rank sum test. p-Values adjusted with the Benjamini-Hochberg correction; UNION.down+B/plasma, 0.22; INT.down+B/plasma, 0.22; UNION.down+CD4+ regulatory T cell (Treg), 0.26; INT. down+UNION.down, 0.26; INT.down, 0.26; UNION.down, 0.26.

Similarly, CD4+ TH17 cells, which were overrepresented in the non-responder groups, exhibit context-dependent roles in tumor immunity and may be associated with both unfavorable and favorable outcomes (*Marques et al., 2021*; *Chang, 2019*). In exploring tumor cell signatures linked to ICI response, non-responder attributes were regulated by STAT3 and NFKB1. The STAT3 and NF-κB pathways are crucial for Th17 cell differentiation and T cell activation (*Park et al., 2014*; *Poholek et al., 2020*). Notably, STAT3 activation in lung cancer orchestrates immunosuppressive characteristics by inhibiting T cell-mediated cytotoxicity (*Jing et al., 2020*). The combined influence of the Th17/STAT3 axis and TRM cell activity in predicting ICI response underscores the complexity of these pathways and suggests that their roles in tumor immunity and therapy response warrants further investigation.

In search of tumor cell signatures associated with the response to ICI, we adopted two approaches, an individual gene level comparison between the responder and non-responder group and a feature extraction approach to decompose data using the NMF and PCA. Both approaches highlighted attributes of non-responders governed by key transcription factors, which play a significant role in immune response regulation. The ability to predict ICI response based on tumor signatures was as accurate as predictions based on immune cell behavior. Integrating data from both immune and tumor cells enhanced the discriminative power (AUC) for identifying responder, suggesting the presence of both interactive and distinct mechanisms of resistance.

Our study has limitations. Primarily, most samples were obtained from metastatic lymph nodes rather than original tumor tissues, potentially not reflecting the tumor microenvironment accurately. However, prior study (*Kim et al., 2020*) and *Figure 1—figure supplement 1* have shown that the immune microenvironment within metastatic lymph nodes closely resembles that of lung tumor tissues, rather than normal lymph nodes. On a positive note, our findings indicate that the immune landscape of metastatic lymph nodes can predict ICI response. Another challenge is the small sample size and the issue of gene expression drop-out, necessitating further studies with a larger patient cohort. Despite these limitations, our study stands out by employing high-throughput scRNA-seq to both tumor cells and immune microenvironment, offering a comprehensive analysis of the multicellular factors that affect ICI response in patients with advanced NSCLC.

## Materials and methods

### Human specimens

This study was approved by the Institutional Review Board (IRB) of Samsung Medical Center (IRB no. 2010-04-039-052). Informed written consent was obtained from all patients enrolled in the study. The study participants included 26 patients diagnosed with lung cancer (*Supplementary file 1*). The study population (n=26) has been treated with the investigator's choice either as a clinical trial (n=5) or as standard clinical practice (n=21). Regardless of the treatment selection, the specimens were prospectively collected based on the study protocol. A total of 33 samples were collected and immediately transferred on ice for tissue preparation. Metastatic lymph nodes, metastatic liver tissues, and lung/bronchus tumor tissues from patients with lung cancer were collected using endobronchial ultrasound bronchoscopy, neck lymph node ultrasound and biopsy, liver biopsy, and percutaneous transthoracic cutting needle biopsy. Tumors, normal lungs, normal lymph nodes, and normal brain tissues were obtained during resection surgery. Pleural fluid was collected from patients with malignant pleural effusion.

### Clinical outcomes

The clinical outcomes of ICI were evaluated based on the RECIST 1.1 (*Eisenhauer et al., 2009*). In this study, we described non-responders as patients with an SD or a PD. Responders were considered as patients with a PR. None of the patients showed a complete response.

### Sample preparation

Single-cell isolation was performed differently depending on the samples. (1) Biopsy samples and nLN were chopped into 2–4 mm pieces and dissociated in an enzyme solution containing collagenase/hyaluronidase (STEMCELL Technologies, Vancouver, Canada) and DNase I, RNase-Free (lyophilized) (QIAGEN, Hilden, Germany) at 37°C for 1 hr. Tissue pieces were re-mixed by gentle pipetting at 20 min intervals during incubation. (2) Tumor and nLung dissociation was performed using a tumor dissociation kit (Miltenyi Biotech, Germany) following the manufacturer's instructions. Briefly, tissue was cut into 2–4 mm pieces and transferred to a C tube containing the enzyme mix (enzymes H, R, and A in RPMI1640 medium). The GentleMACS programs h_tumor_01, h_tumor_02, and h_tumor_02 were run with two 30 min incubations on a MACSmix tube rotator at 37°C. (3) Brain tissue was chopped into 2–4 mm pieces and incubated in an enzyme solution (collagenase [Gibco, Waltham, MA, USA], DNase I [Roche, Basel, Switzerland], and Dispase I [Gibco] in DMEM) at 37°C for 1 hr. Tissue pieces were re-mixed by gentle pipetting at 15 min intervals during incubation. (4) Pleural fluids were transferred to a 50 ml tube, and the cells were spun down at 300×*g*.

Each cell suspension was transferred to a new 50 ml (15 ml for biopsy samples) tube through a 70 µm strainer. The volume in the tube was readjusted to 50 ml (or 15 ml) with RPMI1640 medium, and spun down to remove the enzymes. The supernatant was aspirated, the cell pellet was resuspended in 4 ml of RPMI1640 medium, and dead cells were removed using Ficoll-Paque PLUS (GE Healthcare, Chicago, IL, USA) separation.

For samples subjected to multiplexing, dissociated cells were cryopreserved in CELLBANKER1 (Zenogen, Fukushima, Japan) and thawed for pooling.

### scRNA-seq and read processing

Single-cell suspensions were loaded into a Chromium system (10x Genomics, Pleasanton, CA, USA). Following the manufacturer's instructions, 3' scRNA-seq libraries for the 14 samples were generated

using Chromium Single Cell 3′ v2 Reagent Kits. The 3′ library preparation for EBUS_119 used Chromium Single Cell 3′v3 Reagent Kits. The 5′ scRNA-seq libraries for 12 individual and 2 pooled samples were generated using Chromium Single Cell 5′ v2 Reagent Kit. Libraries were then sequenced on an Illumina HiSeq 2500 for 3′ scRNA-seq and an Illumina NovaSeq 6000 for 5′ scRNA-seq. Sequencing reads were mapped to the GRCh38 human reference genome using Cell Ranger toolkit (v5.0.0).

## SNP genotyping array

Genomic DNA was extracted from the peripheral blood of six patients and subjected to sample multiplexing (DNeasy Blood & Tissue Kit, QIAGEN). The 766,221 single nucleotide polymorphisms (SNPs) were genotyped using Illumina Global Screening Array MG v2, following the manufacturer's instructions. Normalized signal intensity and genotype were processed using Illumina's GenomeStudio v.2 software.

## Demultiplexing of pooled samples

The individuals in sample multiplexing were assigned by a software tool *freemuxlet*, which is an extension of *demuxlet* v2 (**Kang et al., 2018**; https://github.com/statgen/popscle, **Zhang and Kang, 2021**). First, the popscle tool *dsc-pileup* was run with the bam file generated by Cell Ranger toolkit and reference vcf file. The reference was assembled after a lift-over process with GRCh38 from 1000 Genomes Project phase 1 data and the variant allele frequency in East Asian >0.01 were discarded. Next, *freemuxlet* was used to determine the sample identity with default parameters. The individuals were matched based on the similarity between freemuxlet-annotated genotypes and SNP array-detected genotypes.

## Acquisition of scRNA-seq data from LUAD patients

We obtained raw 3′ scRNA-seq from 43 specimens acquired from 33 LUAD patients including early-stage (tLung) and late-stage (tL/B) lung tumor tissues, mLN, nLung, and nLN (**Kim et al., 2020**). Sequencing reads were mapped to the GRCh38 human reference genome using Cell Ranger toolkit (v5.0.0).

## scRNA-seq data analysis

The raw gene-cell-barcode matrix from Cell Ranger pipeline was processed using Seurat v3.2.2 R package (**Stuart et al., 2019**). Cells were selected using two quality criteria: mitochondrial genes (<20%) and gene count (>200). Cell multiplets predicted by Scrublet (**Wolock et al., 2019**) were filtered out. From the filtered cells, the unique molecular identifier (UMI) count matrix was log-normalized and scaled by z-transform while regressing out the effects of cell-cycle variations for subsequent analysis. For batch correction, we used Harmony v1.0 R package (**Korsunsky et al., 2019**) interfacing with Seurat as the *RunHarmony* function. A total of 2000 variably expressed genes were selected using *FindVariableFeatures* with a parameter selection.method="vst". A subset of PCs was selected based on *ElbowPlot* function. Uniform Manifold Approximation and Projection (UMAP) for dimension reduction and cell clustering was performed using *RunUMAP, FindNeighbors*, and *FindClusters* functions with the selected PCs and resolutions (advanced lung cancer patients [total cells, 33 PCs and resolution = 0.3; CD4+ T cells, 28 PCs and resolution = 0.9; CD8+ T cells, 28 PCs and resolution = 0.9; NK cells, 26 PCs and resolution = 0.3; B/plasma cells, 28 PCs and resolution = 0.3; myeloid cells, 30 PCs and resolution = 1.2], LUAD patients [total cells, 23 PCs and resolution = 0.3; T/NK cells, 24 PCs and resolution = 0.9]). We applied the *FindAllMarkers* function to identify DEGs for each cell cluster. Significance was determined using Wilcoxon rank sum test. Genes were selected according to the following statistical thresholds; log fold change >0.25, p-value<0.01, adjusted p-value (Bonferroni correction)<0.01, and percentage of cells (pct) >0.25. Cell identity was determined by comparing the expression of known marker genes and DEGs for each cluster.

## PCA using the proportion of cell lineages and T/NK cell subsets

PCA was performed for the % proportion of cell lineages and T/NK cell subsets in individual LUAD samples using *prcomp* function of stats v3.6.3 R package. For total cells, the percentages of immune and stromal cells were calculated except for epithelial, cycling, and AMB (ambiguous) cells. For T/NK cells, unknown cells annotated as MT high and AMB cells were excluded.

### In silico classification of CD4+ T, CD8+ T, and NK cells

We characterized CD4+ T, CD8+ T, and NK cell populations by combined analysis of gene and protein expression using Cellular Indexing of Transcriptomes and Epitopes by Sequencing data from primary tumor and normal lungs. Among the cells in clusters annotated as T/NK cells, we identified CD3-expressing cells with CD3D or CD3E or CD3G>0 at the RNA level. CD4 and CD8 positive cells were then identified with a cutoff at 55th percentile of ADT level. NK cells were identified based on the RNA expression level of NK cell markers (*XCL1*, *NCAM1*, *KLRD1*, and *KLRF1*) in CD3 negative cells. The gene expression matrix with cell identity of CD4 positive, CD8 positive, and NK was applied as reference data for supervised cell-type classification using *getFeatureSpace* and *trainModel* functions of scPred v1.9.0 R package (*Alquicira-Hernandez et al., 2019*). Finally, we classified T/NK cells in *Figure 1b* into CD4+ T, CD8+ T, and NK cells using scPred *scPredict* function.

### Analysis of TCR/BCR repertoires in CD4+ T, CD8+ T, and B/plasma cells

The data derived from Cell Ranger pipeline for TCR and BCR sequencing data were processed using scRepertoire v1.2.0 R package (*Borcherding et al., 2020*) in R v4.1.1. We selected contigs that generated alpha-beta chain pairs for TCR and heavy-light chain pairs for BCR for subsequent analysis. We called clonotypes based on V(D)JC genes and CDR3 nucleotide sequence with the parameter clonecall="gene+nt". The set of clone types was classified by total frequency using the parameter cloneTypes defined as Single = 1, Small = 5, Medium = 10, Large = 20, and Hyperexpanded = Inf.

### Scoring of T cell functional features

Scores for T cell functional features were calculated as the mean expression of regulatory (*ICOS*, *FOXP3*, *IKZF2*, *LAYN*, *TNFRSF18*, *CTLA4*, *IL21R*, *BATF*, *CCR8*, *IL2RA*, and *TNFRSF4*) and cytotoxic (*CX3CR1*, *PRF1*, *GZMA*, *GZMB*, *GZMH*, *GNLY*, *KLRG1*, and *NKG7*) genes at the log-normalized level.

### Identification of malignant cells based on inferred CNV from scRNA-seq data

Two computational tools, inferCNV v1.2.1 (RRID:SCR_021140, https://github.com/broadinstitute/inferCNV) and CopyKAT v1.0.5 R packages (*Gao et al., 2021*), were used to infer genomic copy numbers from scRNA-seq. In a run with inferCNV, the UMI count matrix of each tumor sample was loaded into inferCNV *CreateInfercnvObject* function along with cell lineage annotations. The reference (normal) cells were selected as cells annotated with T/NK, B/plasma, myeloid, and mast cells. We maintained the proportion of epithelial cells below 20% in each tumor sample using the expression profiles of the nLung and nLN. Inferred copy number variation (CNV) signals were analyzed using inferCNV *run* function using the parameters: cutoff = 0.1, denoise = TRUE, HMM = TRUE, and HMM_type="i6". The signals were then summarized as standard deviations (s.d.) for all windows and the correlation between the CNV in each cell and the mean of the top 5% cells (*Puram et al., 2017*). Cancer cells showing CNV perturbation (>0.03 s.d. or >0.3 CNV correlation) were classified as malignant cells, otherwise as non-malignant cells. The UMI count matrix of each tumor sample was loaded into CopyKAT *copykat* function along with cell lineage annotations using the following parameters: ngene.chr=3, KS.cut=0.05, and norm.cell.names. Cancer cells predicted as aneuploid cells by CopyKAT were classified as malignant cells. Finally, we identified malignant cells, which are cancer cells classified as malignant cells in either inferCNV or CopyKAT.

### Single-cell DEGs between response groups in malignant cells

A total of 12,975 malignant cells were used to identify DEGs in pairwise comparisons according to responder versus non-responder, PR versus PD, and PR versus SD. Differential expression levels were calculated using Seurat *FindMarkers* function with the Wilcoxon rank sum test. Genes were selected according to the following statistical thresholds: log fold change >0.25, p-value<0.01, adjusted p-value (Bonferroni correction)<0.01, and pct>0.25. We reconstructed DEGs as four groups: INT.up, INT.down, UNION,up, and UNION.down, based on with the INT and UNION of up- or downregulated genes for pairwise comparisons between responder versus non-responder, PR versus PD, and PR versus SD. INT.up and INT.down represent the intersection of up- and downregulated genes in the responder group, respectively. UNION.up and UNION.down represent the union of up- and downregulated genes in the responder group, respectively.

## NMF programs of the malignant cells

The UMI count matrix for malignant cells was loaded into *nmf* function of RcppML v0.5.6 R package. A NMF model was learned with a rank of 30 using all genes. For each of the 30 NMF factors, the top-ranked 50 genes in the NMF score were defined as signatures. RECIST-enriched NMF program consisted of selected factors based on their relative sum of loadings. We aggregated and redefined gene signatures of factors included in each NMF program. The uniqueness of each NMF program for RECIST groups was evaluated as an odds ratio using *fisher.test* function of stats v3.6.3 R package. Annotations of NMF programs were assigned using Metascape (*Zhou et al., 2019*).

## RECIST-specific gene modules in malignant cells

The RECIST-specific gene modules were analyzed with a matrix factorization named scINSIGHT (*Qian et al., 2022*) using log-normalized count and 2000 highly variable genes for each sample. For each module, we selected the 100 genes with the highest coefficients. Combinations of the top 100 genes for modules specific to each RECIST group were defined as module genes. The uniqueness of each module for RECIST groups was evaluated as an odds ratio using *fisher.test* function of stats v3.6.3 R package. Annotations of gene modules were assigned using Metascape (*Zhou et al., 2019*).

## PC signatures of the malignant cells

The UMI count matrix for malignant cells was log-normalized and scaled by z-transform while regressing out the effects of cell-cycle variations for PCA. A total of 2000 variably expressed genes selected using *FindVariableFeatures* with selection.method="vst" were used for PCA. PCs were calculated by Seurat *RunPCA* function. PC signatures were selected for 30 genes with + (pos) and – (neg) scores that highly contributed to each PC from PC1 to PC10.

## Functional category analysis

Functional categories representing the enriched gene expression in comparisons for responder vs. non-responder, PR vs. PD, and PR vs. SD as well as in the PCs were identified using fgsea v1.12.0 (*Korotkevich et al., 2019*) R package with parameters: minSize = 10, maxSize = 600, and nperm = 10,000. Gene sets for Gene Ontology (GO) Biological Process were collected from the MSigDB database using msigdbr v7.1.1 R package (*Liberzon et al., 2011*; *Subramanian et al., 2005*). The gene list was ranked by the log fold change for each comparison and feature loadings for each PC. Significant GO terms were selected after collapsing redundant terms using fgsea *collapsePathways* function with a statistical threshold for Benjamini-Hochberg adjusted p-value<0.05.

## Multivariate overall survival analysis of tumor signatures

To evaluate the prognostic potential of tumor signatures, RNA-seq data from LUAD and LUSC samples were retrieved from TCGA data portal (https://portal.gdc.cancer.gov/; *Grossman et al., 2016*). The dataset comprised of 533 primary tumors from TCGA LUAD and 502 primary tumors from TCGA LUSC. Gene expression levels were quantified as (log2 FPKM-UQ+1), where FPKM-UQ represents the upper quartile fragments per kilobase per million mapped reads for each sample.

Clinical variables were first categorized, including age (below and above median age), gender (male and female), pathological stage (I/II and III/IV), distant metastasis (M0 and M1), and nodal status (N0 and Ns). Samples were then classified into high and low groups based on the upper quantile of the mean expression for each tumor signature group. Multivariate survival analysis was conducted using the *analyse_multivariate* function of the survivalAnalysis R package.

## Evaluation of discriminative power of identifying responders for tumor signatures and combinatorial indexes

Classification models of responders and non-responders for PC signatures and combinatorial indexes between tumor and/or immune cells were generated based on in-sample performance and tested by receiver operating characteristic (ROC) curve. Relative numbers between the observed and expected cells (*Ro/e*) for each sample were obtained from the chi-square test (*Guo et al., 2018*). To describe the separability, AUC was calculated using *ROC* function of Epi v2.44 R package with *Ro/e* scores as input. Significance was calculated by Wilcoxon rank sum test and confirmed by adjusting with the Benjamini-Hochberg correction.

## Univariate regression analysis for ICI response

Univariate regression was performed using the *lm* function of stats v3.6.3 with *Ro/e* scores for each sample as input. We evaluated the relationship between the target variable ICI response, classified as responders and non-responders, and one predictor variable of immune cell types, tumor signatures, and clinical factors such as tissue origin, cancer subtype, pathological stage, and smoking status. The significance of predictor was calculated using *Anova* function of car v3.0-9 R package.

## Validation of PC signatures in melanoma cohorts

We used *Riaz et al., 2017*, and *Van Allen et al., 2015*, RNA-seq data from melanoma patients receiving PD-1 and CTLA-4 immune checkpoint therapy to assess expressional changes of PC signatures along RECIST. The mean expression of each PC signature in each RECIST group was calculated as the log2 normalized level.

## Acknowledgements

This study was supported by the Collaborative Genome Program for Fostering New Post-Genome Industry (NRF-2017M3C9A6044633 and NRF-2017M3C9A6044636), Mid-Career Researcher Program (NRF-2022R1A2C1091451), and Basic Research Laboratory Program (RS-2023-00220840) of the National Research Foundation of Korea funded by the Korea government. We also acknowledge the Basic Medical Science Facilitation Program, through the Catholic Medical Center of the Catholic University of Korea funded by the Catholic Education Foundation and the KREONET/GLORIAD service provided by KISTI (Korea Institute of Science and Technology Information).

## Additional information

### Funding

| Funder | Grant reference number | Author |
| --- | --- | --- |
| National Research Foundation of Korea | NRF-2017M3C9A6044633 | Myung-Ju Ahn |
| National Research Foundation of Korea | NRF-2017M3C9A6044636 | Hae-Ock Lee |
| National Research Foundation of Korea | NRF-2022R1A2C1091451 | Hae-Ock Lee |
| National Research Foundation of Korea | RS-2023-00220840 | Hae-Ock Lee |

The funders had no role in study design, data collection and interpretation, or the decision to submit the work for publication.

### Author contributions

Nayoung Kim, Data curation, Formal analysis, Investigation, Visualization, Methodology, Writing – original draft, Writing – review and editing; Sehhoon Park, Resources, Investigation, Writing – original draft, Writing – review and editing; Areum Jo, Hye Hyeon Eum, Dasom Jeong, Minsu Na, Huiram Kang, Investigation, Methodology; Hong Kwan Kim, Kyungjong Lee, Jong Ho Cho, Hyun Ae Jung, Jong-Mu Sun, Se-Hoon Lee, Jin Seok Ahn, Jung-Il Lee, Jung Won Choi, Resources, Investigation; Bo Mi Ku, Resources, Investigation, Methodology; Jeong Yeon Kim, Jung Kyoon Choi, Investigation; Hae-Ock Lee, Conceptualization, Data curation, Supervision, Funding acquisition, Validation, Investigation, Methodology, Writing – original draft, Project administration, Writing – review and editing; Myung-Ju Ahn, Conceptualization, Resources, Data curation, Funding acquisition, Investigation, Writing – review and editing

### Author ORCIDs

Nayoung Kim http://orcid.org/0000-0003-3202-750X
Hae-Ock Lee https://orcid.org/0000-0001-5123-0322

### Ethics

Human subjects: This study was approved by the Institutional Review Board (IRB) of Samsung Medical Center (IRB no. 2010-04-039-052). Informed written consent was obtained from all patients enrolled in the study.

Reviewer #1 (Public review): https://doi.org/10.7554/eLife.98366.3.sa1
Reviewer #2 (Public review): https://doi.org/10.7554/eLife.98366.3.sa2
Author response https://doi.org/10.7554/eLife.98366.3.sa3

---

## Additional files

### Supplementary files

- Supplementary file 1. Patient information of lung cancer immune checkpoint inhibitor (ICI) cohorts.
- Supplementary file 2. List of genes specific to the cell clusters in total cells and each cell lineage.
- Supplementary file 3. List of differentially expressed genes (DEGs) in comparison between immune checkpoint inhibitor (ICI) response groups.
- Supplementary file 4. Details of Gene Ontology (GO) terms significantly enriched in comparisons for immune checkpoint inhibitor (ICI) response groups and principal components (PCs).
- Supplementary file 5. Gene list of tumor signatures.
- MDAR checklist

### Data availability

Raw single-cell RNA sequencing data have been deposited in EGA under accession code EGAD00001008703. Processed data can be accessed from GEO under accession code GSE205335.

The following datasets were generated:

| Author(s) | Year | Dataset title | Dataset URL | Database and Identifier |
|---|---|---|---|---|
| Lee H, Ahn M | 2024 | Single-cell transcriptome profiles of tumor tissues from lung cancer patients receiving immune checkpoint inhibitors | https://www.ncbi.nlm.nih.gov/geo/query/acc.cgi?acc=GSE205335 | NCBI Gene Expression Omnibus, GSE205335 |
| Lee H, Ahn M | 2024 | Single cell RNA sequencing of pre-treatment tissues from lung cancer patients receiving immunotherapy | https://ega-archive.org/datasets/EGAD00001008703 | European Genome-phenome Archive, EGAD00001008703 |

The following previously published datasets were used:

| Author(s) | Year | Dataset title | Dataset URL | Database and Identifier |
|---|---|---|---|---|
| Lee H, Ahn M | 2020 | Comprehensive single cell study of lung adenocarcinoma from early to metastatic stages | https://ega-archive.org/datasets/EGAD00001005054 | European Genome-phenome Archive, EGAD00001005054 |
| Riaz N, Have JJ, Makarov V, Desrichard A, Chan TA | 2018 | Molecular portraits of tumor mutational and micro-environmental sculpting by immune checkpoint blockade therapy | https://www.ncbi.nlm.nih.gov/geo/query/acc.cgi?acc=GSE91061 | NCBI Gene Expression Omnibus, GSE91061 |
| Garraway L, Lander E, Gabriel S | 2016 | Melanoma Genome Sequencing Project | https://www.ncbi.nlm.nih.gov/projects/gap/cgi-bin/study.cgi?study_id=phs000452.v2.p1 | NCBI dbGaP, phs000452.v2.p1 |

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
