## [Editor Report · eLife Assessment]

The authors utilized single-cell RNA-seq profiling of non-small cell lung cancer (NSCLC) patient tumor samples to generate **useful** insights into the determinants of immune checkpoint inhibitor (ICI) responsiveness in NSCLC patients. While some of the findings add weight to the current literature, the analysis is **incomplete** due to the small cohort size and heterogeneous population which has limited their ability to draw statistically supported conclusion after adjusting for multiple hypothesis testing, as well as the lack of functional characterization of the findings. This study would benefit from external cohorts to both validate the findings and justify the statistical analysis undertaken.

---

## [Referee Report · Reviewer #1 (Public review)]

Summary:

The authors study the variability of patient response of NSCLC patients on immune checkpoint inhibitors using single-cell RNA sequencing in a cohort of 26 patients and 33 samples (primary and metastatic sites), mainly focusing on 11 patients and 14 samples for association analyses, to understand the variability of patient response based on immune cell fractions and tumor cell expression patterns. The authors find immune cell fraction and clonal expansion differences, as well as tumor expression differences between responders and non-responders, partly validating previous hypotheses, and partly suggesting new markers for ICI response. Integrating immune and tumor sources of signal the authors claim to improve prediction of response markedly, albeit in a small cohort and using in-sample metrics.

Strengths:

- The problem of studying the tumor microenvironment, as well as the interplay between tumor and immune features is important and interesting and needed to explain heterogeneity of patient response and be able to predict it.

- Extensive analysis of the scRNAseq data with respect to immune and tumor features on different axes of hypothesis relating to immune response and tumor immune evasion using state of the art methods.

- The authors provide an interesting scRNAseq data set with well-curated cell types linked to outcomes data, which is valuable

- High-quality immune cell type annotation including annotations based on additional ADT data

- Integration of TCRseq to confirm subtype of T-cell annotation and clonality analysis

- Interesting analysis of cell programs/states of the (predicted) tumor cells and characterization thereof

Weaknesses:

- Generally a very heterogeneous and small cohort where adjustments for confounding is hard. Additionally, there are many tests for association with outcome, where necessary multiple testing adjustments negate signal and confirmation bias likely, so biological take-aways have to be questioned.

- The authors claim a very high "accuracy" performance, however given the small cohort and possible overfitting due to in-sample ROC the generalization of this to other cohorts is questionable.

- Due to the small cohort with a lot of variability, more external validation is needed to be convincingly reproducible, especially when talking about AUC/accuracy of a predictor.

---

## [Referee Report · Reviewer #2 (Public review)]

Summary:

The authors have utilised deep profiling methods to generate deeper insights into the features of the TME that drive responsiveness to PD-1 therapy in NSCLC.

Strengths:

The main strengths of this work lie in the methodology of integrating single cell sequencing, genetic data and TCRseq data to generate hypotheses regarding determinants of IO responsiveness.

Some of the findings in this study are not surprising and well precedented eg. association of Treg, STAT3 and NFkB with ICI resistance and CD8+ activation in ICI responders and thus act as an additional dataset to add weight to this prior body of evidence. Whilst the role of Th17 in PD-1 resistance has been previously reported (eg. Cancer Immunol Immunother 2023 Apr;72(4):1047-1058, Cancer Immunol Immunother 2024 Feb 13;73(3):47, Nat Commun. 2021; 12: 2606) these studies have used non-clinical models or peripheral blood readouts. Here the authors have supplemented current knowledge by characterization of the TME of the tumor itself.

Weaknesses:

Unfortunately, the study is hampered by the small sample size and heterogeneous population and whilst the authors have attempted to bring in an additional dataset to demonstrate robustness of their approach, the small sample size has limited their ability to draw statistically supported conclusions. There is also limited validation of signatures/methods in independent cohorts and no functional characterisation of the findings. Because of these factors, this work (as it stands) does have value to the field but will likely have a relatively low overall impact.

---

## [Author Response]

The following is the authors’ response to the original reviews.

**Reviewer #1 (Public Review):**
Summary:The authors study the variability of patient response of NSCLC patients on immune checkpoint inhibitors using single-cell RNA sequencing in a cohort of 26 patients and 33 samples (primary and metastatic sites), mainly focusing on 11 patients and 14 samples for association analyses, to understand the variability of patient response based on immune cell fractions and tumor cell expression patterns. The authors find immune cell fraction, clonal expansion differences, and tumor expression differences between responders and non-responders. Integrating immune and tumor sources of signal the authors claim to improve prediction of response markedly, albeit in a small cohort.Strengths:The problem of studying the tumor microenvironment, as well as the interplay between tumor and immune features is important and interesting and needed to explain the heterogeneity of patient response and be able to predict it.Extensive analysis of the scRNAseq data with respect to immune and tumor features on different axes of hypothesis relating to immune response and tumor immune evasion using state-of-the-art methods.The authors provide an interesting scRNAseq data set linked to outcomes data.Integration of TCRseq to confirm subtype of T-cell annotation and clonality analysis.Interesting analysis of cell programs/states of the (predicted) tumor cells and characterization thereof.Weaknesses:Generally, a very heterogeneous and small cohort where adjustments for confounding are hard. Additionally, there are many tests for association with outcome, where necessary multiple testing adjustments would negate signal and confirmation bias likely, so biological takeaways have to be questioned.

Thank you for your comment. We made multiple testing adjustments as suggested in “Recommendations for Authors.”

RNAseq is heavily influenced by the tissue of origin (both cell type and expression), so the association with the outcome can be confounded. The authors try to argue that lymph node T-cell and NK content are similar, but a quantitative test on that would be helpful.

Following the reviewer’s suggestion, we performed principal component analysis (PCA) to assess the influence of tissue of origin on immune and stromal cell populations. In the revised Figure S1g, we quantified the similarity using Euclidean distances of centroids between sample groups based on their tissue of origin in the PC1 and PC3 plot.

The authors claim a very high "accuracy" performance, however, given the small cohort and lack of information on the exact evaluation it is not clear if this just amounts to overfitting the data.

We acknowledge the concern about the high “accuracy” potentially indicating overfitting. To address this, we revised the manuscript to clarify the use of 'accuracy,' 'AUC,' and 'performance' with clearer expressions in the following sections: Abstract (Line 57), Results (Line 264), Discussion (Lines 320-321), Methods (Lines 546-547), Legends for Figure 5c and Figure S8b.

Especially for tumor cell program/state analysis the specificity to the setting of ICIs is not clear and could be prognostic.

Thank you for your comments. As outlined in the ‘Table 2 in the revised manuscript’, we conducted a multivariate survival analysis of tumor signature candidates using the TCGA lung adenocarcinoma (LUAD, n = 533) and squamous cell carcinoma (LUSC, n = 502) cohorts to evaluate their prognostic potential. No tumor cell programs or states were found to be associated with overall survival in either LUAD or LUSC. We added descriptions related to Table 2 in the Results (Lines 249-251) and Methods (Lines 530-542) section.

Due to the small cohort with a lot of variability, more external validation is needed to be convincingly reproducible, especially when talking about AUC/accuracy of a predictor.

Expanding the cohort size was difficult due to limited resources. We recognize the challenges posed by the small and heterogeneous cohort. We have acknowledged these limitations and applied statistical corrections to address them.

**Reviewer #2 (Public Review):**
Summary:The authors have utilised deep profiling methods to generate deeper insights into the features of the TME that drive responsiveness to PD-1 therapy in NSCLC.Strengths:The main strengths of this work lie in the methodology of integrating single-cell sequencing, genetic data, and TCRseq data to generate hypotheses regarding determinants of IO responsiveness.Some of the findings in this study are not surprising and well precedented eg. association of Treg, STAT3, and NFkB with ICI resistance and CD8+ activation in ICI responders and thus act as an additional dataset to add weight to this prior body of evidence. Whilst the role of Th17 in PD-1 resistance has been previously reported (eg. Cancer Immunol Immunother 2023 Apr;72(4):1047-1058, Cancer Immunol Immunother 2024 Feb 13;73(3):47, Nat Commun. 2021; 12: 2606) these studies have used non-clinical models or peripheral blood readouts. Here the authors have supplemented current knowledge by characterization of the TME of the tumor itself.Weaknesses:Unfortunately, the study is hampered by the small sample size and heterogeneous population and whilst the authors have attempted to bring in an additional dataset to demonstrate the robustness of their approach, the small sample size has limited their ability to draw statistically supported conclusions. There is also limited validation of signatures/methods in independent cohorts, no functional characterization of the findings, and the discussion section does not include discussion around the relevance/interpretation of key findings that were highlighted in the abstract (eg. role of Th17, TRM, STAT3, and NFKb). Because of these factors, this work (as it stands) does have value to the field but will likely have a relatively low overall impact.

We acknowledge the challenges posed by the small and heterogeneous cohort. To address this, we tempered our claims related to accuracy by applying statistical testing corrections. We also appreciate the feedback on functional characterization and have expanded the discussion in the revised manuscript to include an overview of specific cell populations and genes.

Related to the absence of discussion around prior TRM findings, the association between TRM involvement in response to IO therapy in this manuscript is counter to what has been previously demonstrated (Cell Rep Med. 2020;1(7):100127, Nat Immunol. 2017;18(8):940-950., J Immunol. 2015;194(7):3475-3486.). However, it should be noted that the authors in this manuscript chose to employ alternative markers of TRM characterisation when defining their clusters and this could indicate a potential rationale for differences in these findings. TRM population is generally characterised through the inclusion of the classical TRM markers CD69 (tissue retention marker) and CD103 (TCR experienced integrin that supports epithelial adhesion), which are both absent from the TRM definition in this study. Additional markers often used are CD44, CXCR6, and CD49a, of which only CXCR6 has been included by the authors. Conversely, the majority of markers used by the authors in the cell type clustering are not specific to TRM (eg. CD6, which is included in the TRM cluster but is expressed at its lowest in cluster 3 which the authors have highlighted as the CD8+ TRM population). Therefore, whilst there is an interesting finding of this particular cell cluster being associated with resistance to ICI, its annotation as a TRM cluster should be interpreted with caution.

Single-cell RNA sequencing (scRNA-seq) can sometimes fail to detect the expression of classical cell type markers due to incomplete capture of a cell’s transcriptome. To determine cell identity, we utilized cell type markers established in previous scRNA-seq studies. In response to your comments, we have added the expression levels of classical TRM markers, including CD69, CD103 (ITGAE), CD44, CXCR6, and CD49a (ITGA1), in the revised Figure 2c. Although these markers were not exclusively expressed in TRM clusters, TRM clusters exhibited relatively high levels of these genes while lacking other clusters’ specific marker genes.

**Reviewer #1 (Recommendations For The Authors):**
General suggestions:When analyzing the association of cell type proportions with outcomes, some adjustment for multiple testing should be considered (either sampling-based, e.g. permutation test, or adjustment based on assumptions of independence of tests, e.g. Bonferroni).

Thank you for your comments. As suggested, we calculated the adjusted p-value using the False Discovery Rate for the association of cell type proportions with outcomes in Figure 3a. The heatmap in Reviewer's ONLY Figure 1, using the adjusted p-value consistently showed the expected grouping of cell types and outcomes. However, the significance did not meet the conventional statistical cutoff criteria. We acknowledge this limitation, which results from statistical testing based on ratio values.

**Author response image 1. sa3fig1:** Heat map with unsupervised hierarchical clustering of proportional changes in cell subtypes within total immune cells. Proportional changes were compared across multiple ICI response groups. The color represents the adjusted -log (p-value) calculated using the False Discovery Rate.

A formal test of clonotype differences (normalized to cell type fraction) would be great as the shown plot 2e could be confounded by cell number and type differences between responders and non-responders.

Thank you for your suggestion. We have revised Figure 2e to display the relative clonotype differences versus CD4+ and CD8+ T cell fractions in each sample. The relative clone size of each cell was calculated by dividing the size of each clone by the total number of CD4+ or CD8+ T cells, respectively.

It could be made a bit more clear when the core group of patients was used (only when associating with outcomes?) and when all other patients were used as well (only cell type annotation?).

As the reviewer correctly noted, we performed scRNA-seq analysis on all specimens, but only the core group of patients was used for the comparative analysis between the responder and non-responder groups. This information has been detailed in the manuscript (Lines 103-105).

For immune cells, it would be interesting to look at expression patterns (NMF, scINSIGHT) as well, not just immune cell fractions and expansion.

In contrast to tumor signatures, immune cell programs are more directly tied to their functional characteristics. Therefore, we focused on annotating immune cells based on their functional properties and conducted comparative analyses between responders and non-responders.

Multiple testing is necessary for the univariate association analysis. Some adjustments for confounders in a multivariate model (despite the size) could be informative.

As shown in ‘Reviewer's ONLY Table 1’, we conducted a multivariate regression analysis of immune and tumor signatures for ICI response, adjusting for clinical variables such as tissue origin, cancer subtype, pathological stage, and smoking status. However, the results were not significant, likely due to the heterogeneity and small size of the cohort.

**Author response table 1. sa3table1:** P-values from univariate and multivariate regression analysis of immune and tumor signatures for ICI response.

		Univariate	Multivariate
p-value <	INT.down	0.009	0.633
0.05 in	UNION.down	0.009	0.346
univariate	CD4+ TH17	0.034	0.429
analysis	PC7.neg	0.047	0.360
	CD4+ Treg	0.048	0.710
p-value <0.05 inmultivariateanalysis	Active NK	0.973	0.004

It is not clear from the manuscript how "accuracy" is measured. The terms "accuracy" and "AUC", as well as "performance" are used interchangeably, a section in the methods with the precise definition is needed.

We have revised the manuscript to clarify the terms 'accuracy,' 'AUC,' and 'performance' by using clearer expressions in the following sections: Abstract (Line 57), Results (Line 264), Discussion (Lines 320-321), Methods (Lines 546-547), Legends for Figure 5c and Figure S8b.

Furthermore, it has to be clear if this is in-sample performance or if there was some train/test split or cross-validation used. Given the small cohort size and wealth of features finding some combination of predictors that could overfit on responders/non-responders would not be surprising.

As the reviewer has noted, we acknowledge the statistical limitations due to the small cohort size. We have revised the sentence on Lines 545-547 “Classification models of responders and non-responders for PC signatures and combinatorial indexes between tumor and/or immune cells were generated based on in-sample performance…”.

Suggestions to improve readability:Line 84: The sentence should be reformulated to improve understanding.

We have revised sentences in lines 81-93.

Line 86: missing a "the".

We have revised the sentences in lines 81-93.

**Reviewer #2 (Recommendations For The Authors):**
"Tumor-infiltrating PD-1 positive T cells have higher capacity of tumor recognition than PD-1 negative T cells" Please look to rephrase this sentence as this is not entirely accurate: PD-1 is upregulated in tumor-experienced T cells as a consequence of antigen recognition ie those cells that recognise tumor will increase PD-1, whereas the sentence as it's currently written indicates that PD1+ cells have an intrinsically increased capacity to kill tumors, which is incorrect.

We have revised the sentence “Tumor-infiltrating PD-1 positive T cells have higher capacity of tumor recognition than PD-1 negative T cells” in lines 86-88 as “More specifically, PD-1 expression is upregulated upon antigen recognition (PMID29296515), indicating that certain T cells in the tumor microenvironment are actively engaged as tumor-specific T cells.” in the revised manuscript.

Cancer subtype abbreviations (eg. SQ, ADC, NUT) are used in figures in the main article and so should be defined in the main text (they are currently only explained in the legend for the supplementary table).

As per the reviewer’s suggestion, the manuscript has been revised to include definitions of cancer type abbreviations in lines 108-110.

Figure S1d-f does not appear to corroborate the statement that "Although there were differences in tissue-specific resident populations, we found that the immune cell profiles, especially T/NK cells of mLN were similar to those of primary tumor tissues indicating the activation of immune responses were 118 consistently observed at metastatic sites (Figure S1d-f)." The diagrams are complex (please explain all abbreviations) and it is not clear how the authors have come to this conclusion. Additionally, cell quantity does not indicate that the 'activation of immune responses' is consistently observed at metastatic sites as these cells could be dysfunctional/bystander.

In the revision, we have quantified the diagrams (Figure S1f) to more clearly highlight the differences in tissue-specific resident populations. We performed principal component analysis (PCA) to evaluate the impact of tissue origin on immune and stromal cell populations. In the revised Figure S1g, we illustrated the quantitative similarity between sample groups using Euclidean distances in the PC plot based on their tissue of origin. Additionally, the legends for Figures S1d and S1e have been updated to include definitions for all abbreviations.

We agree with the reviewer's comment that cell quantity alone may not fully reflect activation of antigen-specific immune responses, even though we annotated the functional T cell subtypes. To better focus on the comparisons of cellular profiles between metastatic sites (mLN) and primary tumors (tLung and tL/B), we removed the sentence “…indicating the activation of immune responses were consistently observed at metastatic sites (Fig. S1d-f).” from the revised manuscript.

In Figure 2c, classical markers for TRM (CD103, CD69) should be included in the description for the definition of the TRM clusters, or their exclusion appropriately explained. The findings regarding the negative correlation between follicular B cells and ICI response are surprising. Figure S3, the cluster identified as Follicular B cells contains MS4A1 (CD20) and HLA-DRA. Classical markers are CD20 (pan-B cell), CD21 (CR2), CD23, and IgD/IgM (double positive), and as such it is not clear if the authors have appropriately annotated this cluster as representing follicular B cells. These classical markers should be included in the interpretation of the cell clustering or their exclusion appropriately explained.

We appreciate your comments. In response, we have added the expression levels of classical TRM markers such as CD69, CD103 (ITGAE), CD44, CXCR6, and CD49a (ITGA1), in the revised Figure 2c. Additionally, we revised the dot plot showing the mean expression of marker genes in each cell cluster for B/Plasma cells (revised Figure S3b) by incorporating classical markers for Follicular B cells, such as CD21 (CR2), CD23 (FCER2), IgD (IGHD), IgM (IGHM).

Figure 2f is rather confusing for the reader. I would recommend changing to an alternative plot that shows logP and response in a different way. If keeping to this plot type please clarify why plotting response vs PD, and whether the lower left quadrant indicates patients with progressive disease and the top right indicates responders as the interpretation is not clear currently.

Thank you for your feedback. To address the concerns raised, we have updated the figure legend for Figure 2f to clarify the interpretation of the quadrants: “The lower left quadrant shows cell types overrepresented in the poor responder groups, while the upper right quadrant indicates cell types overrepresented in the better responder groups”. This clarification aims to help readers understand that the lower left quadrant reflects cell types associated with worse treatment outcomes, while the upper right quadrant reflects cell types associated with improved therapeutic responses.

The terms "PC7.neg, INT.down, and UNION.down" are included in the results with no explanation to the reader of what they are or how to interpret them. The methods description "We constructed DEGs with 470 intersections (INT) and union (UNION) of up- or down-regulated genes for comparisons" does not sufficiently describe how they were generated/calculated and, therefore, this is difficult for the reader to interpret in the final results section. Please add an additional explanation for the reader in the final section of the results/Figure 5 and in the methods.

Following the reviewer’s suggestion, we added additional explanation in the Results section (lines 258-261): “PC7.neg denotes genes negatively correlated with PC7, a principal component extracted from PCA that distinguishes tumor cells in poor response groups. INT.down and UNION.down represent the intersection and union of down-regulated genes in the responder group, respectively.”. We also explained the details in the Methods section (lines 489-495): “We reconstructed DEGs as four groups: INT.up, INT.down, UNION,up, and UNION.down, based on with the intersection (INT) and union (UNION) of up- or down-regulated genes for pairwise comparisons between responder versus non-responder, PR versus PD, and PR versus SD. INT.up and INT.down represent the intersection of up- and down-regulated genes in the responder group, respectively. UNION.up and UNION.down represent the union of up- and down-regulated genes in the responder group, respectively.”

The TRM and Th17+ T cell populations are highlighted in the abstract as being related to ICI resistance, but these populations of cells are not even mentioned in the discussion. Likewise, STAT3 and NFkb pathways are also highlighted in the abstract but absent in the discussion section. Please discuss the relevance of these findings, particularly given the prior studies demonstrating the opposite impact of TRM populations in NSCLC.

We have expanded the discussion in the revised manuscript (Lines 295-313) to address the roles of TRM and Th17+ T cell, as well as the STAT3 and NF-κB pathways, in association with ICI resistance in NSCLC.

“The identification of an abundance of CD4+ TRM cells as a negative predictor of ICI response is an unexpected finding, considering that higher frequencies of TRM cells in lung tumor tissues are generally associated with better clinical outcomes in NSCLC (PMID28628092). This is largely due to their role in sustaining high densities of tumor-infiltrating lymphocytes and promoting anti-tumor responses. Additionally, previous studies have demonstrated that TRM cell subsets coexpressing PD-1 and TIM-3 are relatively enriched in patients who respond to PD-1 inhibitors (PMID31227543). However, recent findings suggest that pre-existing TRM-like cells in lung cancer may promote immune evasion mechanisms, contributing to resistance to immune checkpoint blockade therapies (PMID37086716). These observations suggest that the roles of TRM subsets in tumor immunity are highly context-dependent.

Similarly, CD4+ TH17 cells, which were overrepresented in the non-responder groups, exhibit context-dependent roles in tumor immunity and may be associated with both unfavorable and favorable outcomes (PMID34733609; PMID30941641). In exploring tumor cell signatures linked to ICI response, non-responder attributes were regulated by STAT3 and NFKB1. The STAT3 and NF-κB pathways are crucial for Th17 cell differentiation and T cell activation (PMID24605076; PMID32697822). Notably, STAT3 activation in lung cancer orchestrates immunosuppressive characteristics by inhibiting T-cell mediated cytotoxicity (PMID31848193). The combined influence of the Th17/STAT3 axis and TRM cell activity in predicting ICI response underscores the complexity of these pathways and suggests that their roles in tumor immunity and therapy response warrants further investigation.”